# A survival selection strategy for engineering synthetic binding proteins that specifically recognize post-translationally phosphorylated proteins

Bunyarit Meksiriporn[1,6], Morgan B. Ludwicki[2,6], Erin A. Stephens[3], Allen Jiang[2], Hyeon-Cheol Lee [2], Dujduan Waraho-Zhmayev[4], Lutz Kummer[5], Fabian Brandl[5], Andreas Plückthun [5] & Matthew P. DeLisa [1,2,3]

There is an urgent need for affinity reagents that target phospho-modified sites on individual proteins; however, generating such reagents remains a significant challenge. Here, we describe a genetic selection strategy for routine laboratory isolation of phospho-specific designed ankyrin repeat proteins (DARPins) by linking in vivo affinity capture of a phosphorylated target protein with antibiotic resistance of *Escherichia coli* cells. The assay is validated using an existing panel of DARPins that selectively bind the nonphosphorylated (inactive) form of extracellular signal-regulated kinase 2 (ERK2) or its doubly phosphorylated (active) form (pERK2). We then use the selection to affinity-mature a phospho-specific DARPin without compromising its selectivity for pERK2 over ERK2 and to reprogram the substrate specificity of the same DARPin towards non-cognate ERK2. Collectively, these results establish our genetic selection as a useful and potentially generalizable protein engineering tool for studying phospho-specific binding proteins and customizing their affinity and selectivity.

[1] Nancy E. and Peter C. Meinig School of Biomedical Engineering, Cornell University, Ithaca, NY 14853, USA. [2] Robert F. Smith School of Chemical and Biomolecular Engineering, Cornell University, Ithaca, NY 14853, USA. [3] Biochemistry, Molecular and Cell Biology, Cornell University, Ithaca, NY 14853, USA. [4] Biological Engineering Program, Faculty of Engineering, King Mongkut's University of Technology Thonburi, Bangkok 10140, Thailand. [5] Department of Biochemistry, University of Zürich, 8057 Zürich, Switzerland. [6] These authors contributed equally: Bunyarit Meksiriporn, Morgan B. Ludwicki. Correspondence and requests for materials should be addressed to M.P.D. (email: md255@cornell.edu)

Many cellular activities are controlled by protein post-translational modifications (PTMs), with over 200 different types having been identified[1] including phosphorylation, acetylation, ubiquitination, methylation, and glycosylation. PTMs serve to functionally diversity the proteome by finely tuning the structure, stability, activity, subcellular localization, and protein interaction partners of the modified proteins[2]. Whereas asparagine-linked (N-linked) glycosylation dominates the number of putative PTMs, phosphorylation dominates the number of experimentally confirmed PTMs by an order of magnitude[3]. Indeed, phosphorylation easily ranks as one of the most common PTMs in eukaryotes with well over 100,000 phosphosites identified in humans and related mammals and over two-thirds of the 23,000 proteins encoded by the human genome demonstrated to be covalently modified with phosphate by the collective activity of >500 protein kinases[4–8]. Phosphorylation is particularly important in signal propagation where it regulates the function of numerous proteins in signaling networks by activating or inhibiting enzyme activity through allosteric conformational changes[9–11]. In light of the pivotal role played by phosphorylation in signal transduction, it is hardly surprising that aberrant phosphorylation either directly causes or is a consequence of many human diseases, in particular cancer[12].

Over the last two decades, mass spectrometry-based proteomics has emerged as one of the most effective approaches for analyzing PTMs and identifying their sites of attachment on proteins, including phosphoproteins[7,13–15]. Given the steady increase in the number of functionally important phosphorylation sites that have been uncovered, there is a growing need for phospho-specific binding molecules[16] that can be developed for traditional biochemical approaches as well as advanced techniques such as single-cell analysis[17–19] and high-throughput assay systems[20–22]. The most common affinity reagents for detecting PTMs, and more specifically phospho-modified sites, are conventional monoclonal antibodies (mAbs) that have been raised in mice[23]. However, the use of animal immunization to isolate phospho-specific mAbs is low-throughput, expensive, and time-consuming, and is further hampered by technical challenges associated with the widespread occurrence of phospho-epitopes, which renders them weakly immunogenic in intact immune systems. Hence, for most targets, no specific reagents exist, and in cases where commercially mAbs are available, they are known to be of highly variable quality and limited utility[17,24].

The use of recombinant technologies that take advantage of synthetic antibody libraries have emerged as a viable approach to specifically select for binders against phospho-modified sites on individual targets[25–27]; however, these approaches are often less efficient than immunization. Another challenge is that the resulting antibody fragments require intradomain disulfide bonds for conformational stability, thereby precluding their use as "intrabodies" in the reducing intracellular environment where most phosphoproteins of interest naturally reside. This bottleneck can be overcome by using alternative non-antibody scaffolds for molecular recognition such as designed ankyrin repeat proteins (DARPins), which do not contain disulfide bonds and can be expressed in soluble form with high yields in the cytoplasm of living cells thereby allowing for intracellular applications[28,29]. Indeed, using complex DARPin libraries, Plückthun and coworkers isolated target-specific binders that could reliably differentiate between two states of a protein post-translationally modified by phosphorylation, and were subsequently shown to be functional in the cytoplasm of eukaryotic cells[30]. However, a drawback to the synthetic library approaches reported to date is that they rely on in vitro selection methods, such as phage display or ribosome display, which are technically demanding and labor intensive, and are implemented in cell-free environments that may not accurately reflect the complex conditions inside of a cell.

To address these shortcomings, we sought to adapt a previous genetic assay termed FLI-TRAP (functional ligand-binding identification by Tat-based recognition of associating proteins) for selection of phospho-specific binders directly in living cells in a manner that greatly simplifies the process by which synthetic libraries are interrogated. FLI-TRAP is a complete in vivo selection and evolution technology based on the unique ability of the twin-arginine translocation (Tat) system to efficiently colocalize noncovalent complexes of two folded polypeptides to the Escherichia coli periplasm[31]. This method has proven especially useful for high-throughput selection of single-chain Fv (scFv) antibodies that bind strongly to their cognate protein antigens in the intracellular environment[31–34].

Here, FLI-TRAP was functionally extended for detecting phospho-specific interactions using the extracellular signal-regulated kinase 2 (ERK2), a member of the mitogen-activated protein kinase (MAPK) family, as a model system for specific intracellular targeting of a protein as a function of its post-translational modification. ERK2 activation is mediated by the upstream MAP/ERK kinase 1 (MEK1), which phosphorylates a threonine and tyrosine within a flexible surface loop that undergoes small but significant conformational rearrangements upon modification[11]. Upon combining FLI-TRAP with a reconstituted MAP kinase phosphorylation cascade that promotes cytoplasmic phospho-modification of ERK2[35], the reformatted genetic assay called phospho-FLI-TRAP (hereafter PhLI-TRAP) reliably reported the specificity and selectivity of an existing panel of DARPins[30] that selectively bind the nonphosphorylated (inactive) form of ERK2 or its doubly phosphorylated (active) form, pERK2. Following validation, PhLI-TRAP was successfully used to enhance the affinity of a phospho-specific DARPin for its cognate pERK2 antigen as well as to reprogram the specificity of the same parental DARPin for binding to noncognate ERK2. Importantly, by linking antibiotic resistance with phospho-epitope binding in the cytoplasm of E. coli cells, the PhLI-TRAP method obviates the need for purification or immobilization of the phosphoprotein target and only requires selective plating of bacteria on solid medium to uncover productive binders. Hence, our genetic selection represents a simpler alternative to existing methods, offering savings in time and resources, while at the same time providing a reliable tool for generating phospho-specific affinity reagents that are both high quality and renewable.

## Results

**A genetic selection for phospho-modified proteins.** To develop the PhLI-TRAP method for direct selection of phospho-modified substrate proteins (Fig. 1), we employed DARPins against either the unphosphorylated or the doubly phosphorylated form of the MAPK ERK2 (ERK2 or pERK2, respectively). ERK2 is activated by phosphorylation on Thr183 and Tyr185 residues, which is catalyzed by MEK1[35]. Specifically, DARPin pE59, which is selective for pERK2, was cloned into a plasmid derived from pBAD18[33] that introduced the N-terminal Tat signal peptide derived from trimethylamine N-oxide reductase (spTorA) for targeting to TatABC followed by an RGS-His tag for convenient detection. In parallel, a second plasmid was created in which human ERK2 was genetically fused to the N-terminus of mature TEM-1 β-lactamase (Bla), which acts as a selectable reporter for transport to the periplasm. To generate phosphorylated ERK2 in the cytoplasm, the gene encoding a constitutively active mutant of human MEK1, namely MEK1[R4F], which is capable of activating ERK2 when expressed in E. coli[35], was cloned bicistronically into

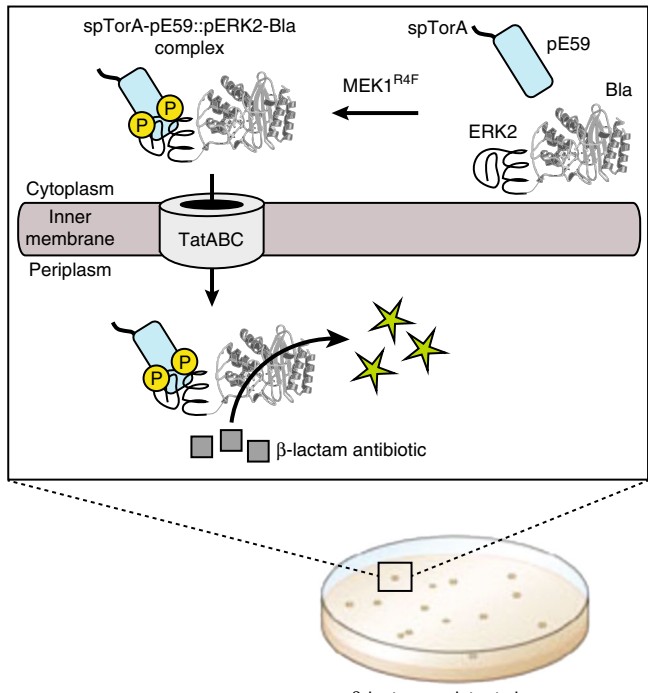

**Fig. 1** PhLI-TRAP-based isolation of phospho-specific binding proteins. Schematic representation of engineered assay for co-translocation of interacting receptor-antigen pairs via the Tat translocase (TatABC). The assay enables discovery and optimization of synthetic binding proteins (e.g., DARPins) with affinity for phospho-modified target antigens simply by demanding bacterial growth on β-lactam antibiotics such as carbenicillin (Carb), without the need for purification or immobilization of the phosphoprotein target. The Tat signal peptide chosen was spTorA, the reporter enzyme was Bla, the synthetic binding protein was an ERK2- or pERK2-specific DARPin, and the antigen was ERK2. Phosphorylation status of ERK2 was toggled by expression of the constitutively active upstream kinase MEK1$^{R4F}$, which doubly phosphorylates (yellow P circles) ERK2 in the cytoplasm of living *E. coli* cells

the low-copy DARPin expression plasmid. We hypothesized that co-expression of spTorA-pE59, ERK2-Bla, and MEK1$^{R4F}$ would result in the formation of a heterodimeric complex between spTorA-pE59 and phosphorylated ERK2-Bla (pERK2-Bla) in the cytoplasm, which would subsequently be cotranslocated to the periplasm according to the "hitchhiker" mechanism (Fig. 1)[36]. Importantly, export of Bla to the periplasm renders *E. coli* cells resistant to β-lactam antibiotics, thereby enabling simple clonal selection to discriminate phospho-specific interactions.

In line with our hypothesis, co-expression of these three constructs in wild-type *E. coli* MC4100 cells resulted in MEK1$^{R4F}$-dependent phosphorylation of ERK2-Bla (Supplementary Fig. 1a) and concomitant cotranslocation of the phospho-modified substrate to the periplasmic space as confirmed by western blot analysis (Supplementary Fig. 1b). We observed no significant translocation of pERK2-Bla when pE59 was replaced with the well characterized DARPin OFF7, which is specific for maltose-binding protein (MBP)[28], confirming the specificity of pE59 for its cognate form of ERK2. Likewise, there was no significant translocation of pERK2-Bla in the presence of: (i) an export-defective mutant, spTorA(KK)-pE59, in which the essential twin-arginine residues of the N-terminal Tat signal peptide were mutated to lysines thereby abolishing export out of the cytoplasm (Supplementary Fig. 1b); or (ii) a double mutant of ERK2 (mERK2) in which the kinase-essential phosphorylation

sites at Thr183 and Tyr185 were mutated to Glu and Phe, respectively (Supplementary Fig. 1c).

When cells that exported pERK2-Bla to the periplasm were analyzed by spot plating analysis, we observed strong carbenicillin (Carb) resistance to a level that was even greater than that observed for positive control cells co-expressing OFF7 with MBP-Bla (Fig. 2 and Supplementary Fig. 2a). In contrast, negative control cells co-expressing a Bla fusion involving the c-Jun N-terminal kinase 2 (JNK2), a MAPK that is highly similar to ERK2, exhibited little to no Carb resistance in the presence or absence of MEK1$^{R4F}$ (Fig. 2 and Supplementary Fig. 2a), consistent with the known specificity for pE59[30]. Importantly, the resistance conferred by pE59, but not OFF7, was dependent on MEK1$^{R4F}$ co-expression, indicating that the selectivity of pE59 for pERK2 over ERK2 was maintained. Along similar lines, we found that the resistance conferred by pE59 was reduced to background in cells co-expressing MEK1$^{R4F}$ and the phospho-mutant mERK2-Bla (Supplementary Fig. 2b), providing further support of phospho-selectivity. Collectively, these results confirm that both the high specificity and selectivity of pE59 for the phosphorylated form of ERK2 was retained in the genetic selection.

**Genetic selection reconstitutes selectivity of other DARPins.** Encouraged by the results with pE59, we next investigated the ability of the genetic selection to recapitulate the selectivity of two ERK2-binding DARPins, E8 and E38, and two ERK2/pERK2-binding DARPins, EpE82 and EpE89, that were all described previously[30]. Spot plating experiments were performed as above but with pE59 replaced by one of these alternative DARPins in the low-copy DARPin expression plasmid. In the case of EpE82 and EpE89, which are known to bind equally well to both forms of ERK2[30], we observed resistance profiles in the presence of MEK1$^{R4F}$ that rivaled the level of resistance observed for cells co-expressing spTorA-pE59 with pERK2-Bla (Fig. 3a and Supplementary Fig. 3). These clones conferred relatively less resistance to cells in the absence of MEK1$^{R4F}$; however, resistance levels were significantly higher than the pERK2-specific clone pE59, confirming the ability of these clones to recognize both ERK2 forms. In the case of E8 and E38, strong resistance was only observed in the absence of MEK1$^{R4F}$ (Fig. 3b and Supplementary Fig. 3), consistent with the selectivity of these two DARPins for the nonphosphorylated form of ERK2.

**Selection of pE59 variants with improved affinity for pERK2.** Our previous studies confirmed that the efficiency with which a TatABC-targeted binding protein escorts its cognate antigen-Bla fusion to the periplasm depends on both the expression/stability of the binding protein in vivo and its affinity for the antigen[31,33]. Since most DARPins including those described above have naturally high soluble expression yields in the *E. coli* cytoplasm[37], we hypothesized that pE59 variants with enhanced affinity for cognate pERK2 antigen could be readily isolated by simply demanding cell growth on Carb concentrations that would otherwise inhibit the growth of cells expressing the parental pE59 clone. To test this hypothesis, we generated an error-prone PCR library of pE59 sequences and cloned these just after the spTorA signal peptide in the low-copy expression plasmid (that also included the gene encoding MEK1$^{R4F}$). Following co-transformation of wild-type MC4100 cells with the plasmid library along with the ERK2-Bla plasmid, positive clones were selected on high concentrations of Carb (300-500 μg/mL) from a starting library that contained ~$10^5$–$10^6$ members. These Carb concentrations were chosen because they supported outgrowth of positive hits from the library but inhibited outgrowth of individual cells expressing the parental pE59 sequence. Following a

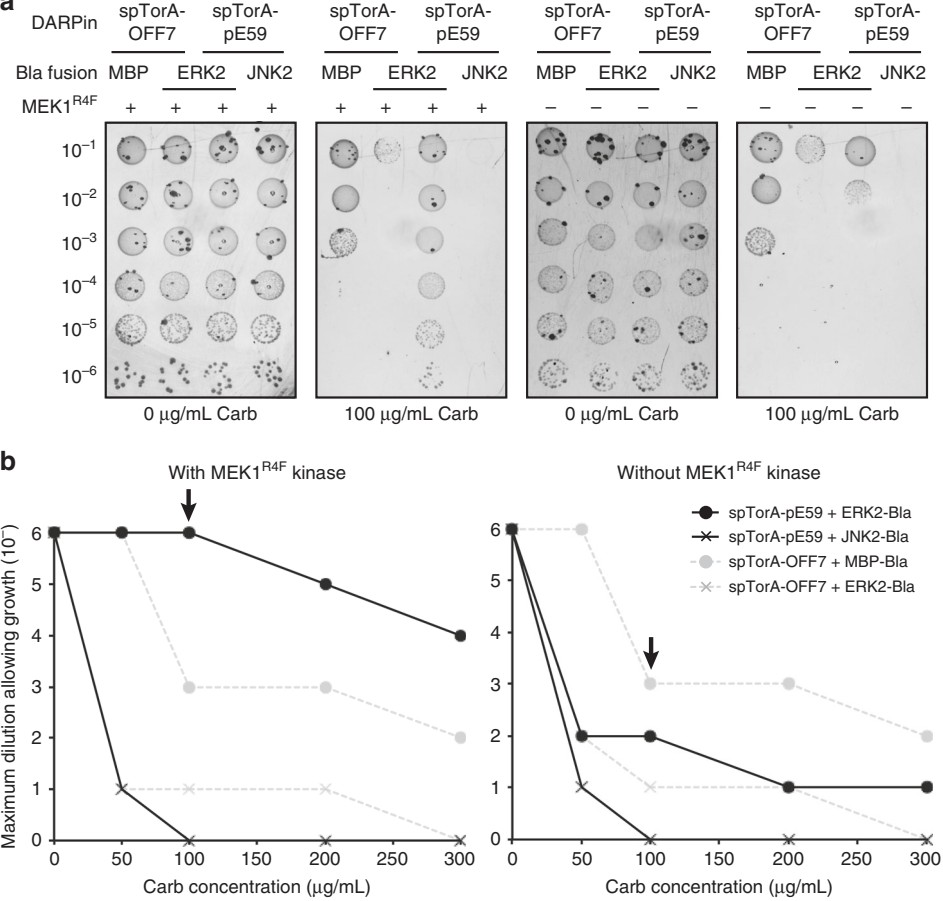

**Fig. 2** Phenotypic selection of pERK2-specific DARPin binding. **a** Representative spot titer images and **b** survival curves for serially diluted *E. coli* MC4100 cells co-expressing TatABC along with the antigen-Bla reporter fusion (MBP-Bla, ERK2-Bla, or JNK2-Bla) and a Tat-targeted DARPin (spTorA-pE59 or spTorA-OFF7) as indicated. Resistance of cells was evaluated in the presence (+) or absence (−) of MEK1$^{R4F}$ kinase. Overnight cultures were serially diluted in liquid LB and plated on LB agar supplemented with Carb. Maximal cell dilution that allowed growth is plotted versus Carb concentration. Arrow in **b** indicates Carb concentration that is depicted in image panel **a** above the graphs and corresponds to 100 µg/ml Carb. Source data for this figure is available in the Source Data File

single round of survival-based enrichment using the PhLI-TRAP assay, 10 putative hits were randomly chosen from selective plates for characterization. Sequencing revealed that one of these was a false positive, and that two of the remaining nine were isogenic duplicates. To confirm that the greater resistance conferred by the eight unique clones was due to mutations in pE59 and not elsewhere in the plasmid, all isolated DARPin sequences were back-cloned into the original low-copy vector and used to transform wild-type MC4100 cells carrying the ERK2-Bla plasmid. Spot plating of cells co-expressing the back-cloned genes along with ERK2-Bla and MEK1$^{R4F}$ confirmed that all eight positive hits conferred significantly greater Carb resistance to cells compared to that conferred by parental pE59. Three of these in particular, clones pEM1 (selected on 400 µg/mL Carb), pEM2 (selected on 300 µg/mL Carb), and pEM3 (selected on 400 µg/mL Carb), showed very strong resistance phenotypes (Fig. 4a and Supplementary Fig. 4) and were chosen for further analysis.

Next, we evaluated binding activity of the isolated clones by indirect enzyme-linked immunosorbent analysis (ELISA) using immobilized ERK2 and pERK2 as antigens. In accordance with the drug resistance results, pEM1, pEM2, and pEM3 all exhibited significantly higher binding activity against pERK2 compared to pE59 with clone pEM1 showing the greatest improvement (Fig. 4b). When the same clones were assayed for binding against ERK2, all showed very low binding activity that was slightly

higher than pE59 (Fig. 4b). To quantify the affinity for the most improved clone, pEM1, the equilibrium dissociation constant $K_D$ was determined for binding to the ERK2 and pERK2 antigens by kinetic surface plasmon resonance (SPR) measurements on a Biacore instrument. Overall, we determined that the stronger binding measured for pEM1 in spot plating and ELISA experiments above resulted from a >five-fold improvement in pERK2 affinity, to 87.1 nM, while the observed selectivity in these assays stemmed from a >40-fold difference in binding affinity to cognate pERK2 versus non-cognate ERK2 (Table 1 and Supplementary Fig. 5). The apparent selectivities for both pE59 and pEM1 may be even higher because SPR signals for the non-cognate ERK2 form were very low and thus led to an imprecise estimation of $K_D$. This could also explain in part the discrepancy in the selectivity values for pE59 reported here and by Kummer et al.[30]. Taken together, these results suggest that by performing genetic selections in a MEK1$^{R4F}$-expressing strain background, the selectivity of the affinity-matured pE59 variants was not compromised and remained strongly biased towards phospho-modified ERK2.

Sequencing of the three hits revealed that a relatively small number of amino acid changes (F67Y in pEM1, L7M and D60G in pEM2, and L55V, N62K, and I83V in pEM3) is responsible for the increased binding affinity. Collectively, the mutations primarily mapped to the ankyrin repeat modules between the

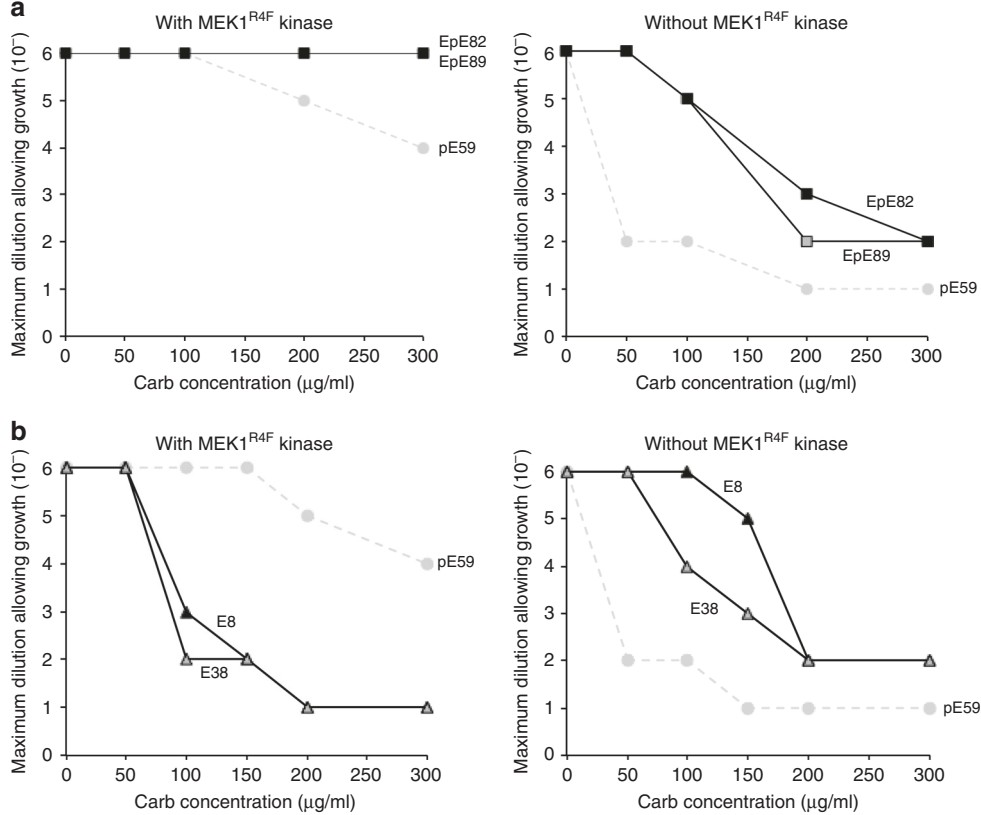

**Fig. 3** Phenotypic selection of DARPins binding cognate antigens. Survival curves for serially diluted *E. coli* MC4100 cells co-expressing TatABC along with ERK2-Bla and either: **a** Tat-targeted DARPins EpE82 (black squares) and EpE89 (gray squares) that recognize both ERK2 forms; or **b** Tat-targeted DARPins E8 (black triangles) and E38 (gray triangles) that specifically recognize nonphosphorylated ERK2. Resistance of cells was evaluated in the presence (left) or absence (right) of MEK1$^{R4F}$ kinase as indicated. Overnight cultures were serially diluted in liquid LB and plated on LB agar supplemented with Carb. Maximal cell dilution that allowed growth is plotted versus Carb concentration. Resistance profiles for pERK2-specific DARPin pE59 are shown (gray circles) for comparison. Source data for this figure is available in the Source Data File

N- and C-terminal capping repeats (Supplementary Table 1). Interestingly, the significantly enhanced affinity of pEM1 arises from a single mutation, F67Y, to a residue that directly contacts the surface of pERK2 (Fig. 4c). For clones pEM2 and pEM3, most of the mutations (L55, N62, D60, and I83) appear to be internal to the DARPins and may not contact pERK2 directly (Fig. 4c), although it is possible that these mutations alter the structure of the clustered helices in a way that enhances contact with pERK. In the case of pEM3, an additional L7M mutation in the N-capping repeat might also explain the improved binding as this residue is in position to interact with the surface of pERK2 (Fig. 4c).

**Reprogramming the specificity and selectivity of pE59.** Encouraged by our success isolating pE59 variants with stronger affinity to cognate pERK2, we next attempted to use our genetic selection to redirect pE59 binding to nonphosphorylated ERK2. Similar to the approach outlined above, an error-prone library of pE59 sequences was generated and cloned just after the spTorA signal peptide in the low-copy expression plasmid; however, the gene encoding MEK1$^{R4F}$ was omitted. Following co-transformation of wild-type MC4100 cells with the plasmid library along with the ERK2-Bla plasmid, positive clones were selected on moderate Carb concentrations (50 and 300 μg/mL) from a starting library that contained ~$10^5$–$10^6$ members. At these concentrations, we anticipated that outgrowth of individual cells expressing the parental pE59 sequence would be inhibited,

thereby limiting outgrowth to only positive hits from the library. Following one round of survival-based enrichment using the PhLI-TRAP assay in the absence of MEK1$^{R4F}$-mediated phosphorylation, seven putative hits were randomly chosen from selective plates (EpEM1-EpEM6 selected on 50 μg/mL Carb and EpEM7 on 300 μg/mL Carb; Supplementary Table 1) and subjected to further characterization. Following the same sequencing and back-transformation procedure described above, we determined that all of these hits were unique sequences that conferred a true positive phenotype in the PhLI-TRAP assay. Specifically, spot plating of cells co-expressing back-cloned genes along with ERK2-Bla in the absence of MEK1$^{R4F}$ confirmed that all seven hits conferred greater Carb resistance to cells compared to that conferred by parental pE59 (Fig. 5a and Supplementary Fig. 6). Two clones in particular, EpEM6 and EpEM7, stood out for their high level of Carb resistance, which suggested that each had acquired strong binding activity toward non-cognate ERK2.

To determine whether these newly evolved DARPin variants retained parental binding activity to pERK2, we performed spot plating analysis of cells co-expressing EpEM6 or EpEM7 along with ERK2-Bla and MEK1$^{R4F}$. Indeed, both variants conferred strong resistance to cells expressing pERK2 (Fig. 5b), indicating that reprogramming substrate specificity toward non-cognate ERK2 resulted in the evolution of promiscuous variants that bound both ERK2 forms. Interestingly, whereas EpEM6 behaved similarly to pE59 in the presence of pERK2, EpEM7 conferred a level of resistance that was on par with affinity-matured pEM1 (Fig. 5b). In light of this result, it is interesting to note that two of

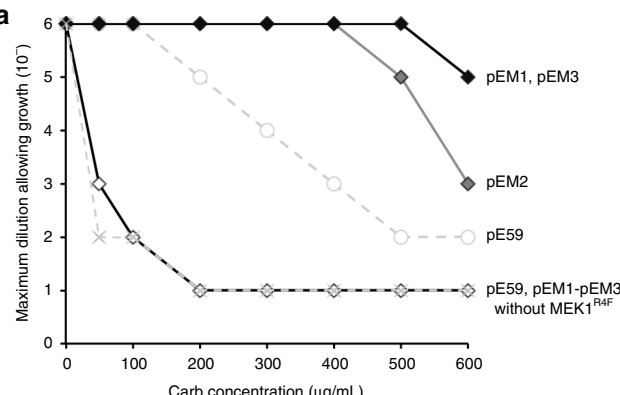

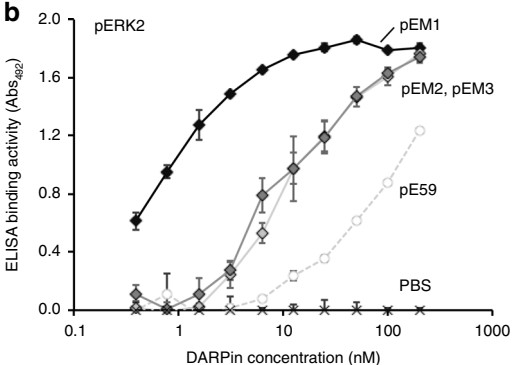

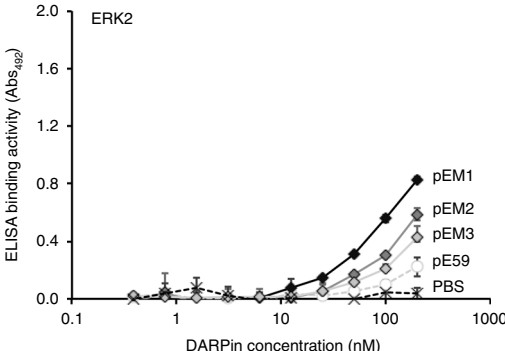

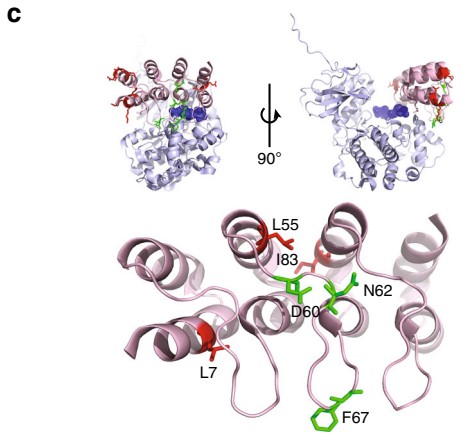

the three mutations in EpEM7, namely L6P and N62Y, are similar to the L7M, D60G, and N62K mutations uncovered in affinity-matured pEM2 and pEM3 (Supplementary Table 1), which might explain the strengthened binding to pERK2 by EpEM7.

To further investigate the selectivity of these two clones, binding to both ERK2 forms was quantified by kinetic SPR

**Fig. 4** Isolation of DARPin variants with enhanced pERK2 affinity. **a** Survival curves for serially diluted *E. coli* MC4100 cells co-expressing TatABC along with ERK2-Bla and Tat-targeted DARPins pEM1 (light gray), pEM2 (dark gray), and pEM3 (black). Resistance of cells was evaluated in the presence (filled diamonds) or absence (empty diamonds) of MEK1[R4F] kinase. Overnight cultures were serially diluted in liquid LB and plated on LB agar supplemented with Carb. Maximal cell dilution that allowed growth is plotted versus Carb concentration. Resistance profiles for pERK2-specific DARPin pE59 are shown with (empty circles) and without (light gray x marks) MEK1[R4F] for comparison. **b** ELISA binding activity for purified DARPins pE59, pEM1, pEM2, and pEM3 against immobilized pERK2 (top) or ERK2 (bottom). PBS served as a negative control. All data are the average of three biological replicates and the error bars represent the standard deviation (SD). **c** Location of mutations in DARPin variants pEM1, pEM2, and pEM3 mapped onto the pE59-pERK2 co-crystal structure. The structure was derived from PDB ID 3ZUV described in Kummer et al.[30], and the schematic was generated using PyMOL software. The DARPin is shown in light red, pERK2 in light blue, the phosphorylated T185 and Y187 active site residues of ERK2 in dark blue spheres, and the mutations found in helix and loop secondary structures in red and green, respectively. Source data for this figure is available in the Source Data File

### Table 1 Affinity and selectivity of selected DARPins for ERK2 and pERK2

| DARPin | $K_D$ (M), pERK2 | $K_D$ (M), ERK2 | Selectivity |
|---|---|---|---|
| pE59 | $453 \times 10^{-9}$ | $>3.5 \times 10^{-6}$ | >7.7 |
| pEM1 | $87.1 \times 10^{-9}$ | $>3.5 \times 10^{-6}$ | >40 |
| EpEM6 | $633 \times 10^{-9}$ | $784 \times 10^{-9}$ | 1.0 |
| EpEM7 | $25.2 \times 10^{-9}$ | $222 \times 10^{-9}$ | 8.8 |

measurements. In agreement with the resistance profiles seen above, clones EpEM6 and EpEM7 both exhibited dramatically enhanced affinity for non-cognate ERK2 compared to parental pE59, with measured $K_D$ values in the 200–800 nM range (Table 1 and Supplementary Fig. 7). Because the affinity of EpEM6 for pERK2 remained unchanged, the selectivity of this DARPin variant was significantly relaxed compared to the pERK2-biased pE59 from which it was derived (selectivity of ~1 fold versus ~8 fold, respectively). In contrast, clone EpEM7 retained high selectivity for cognate pERK2 (~9 fold) due in large part to the unexpected acquisition of stronger affinity for the pERK2 form during the library selection process.

## Discussion

In this study, we developed a genetic selection strategy called PhLI-TRAP that enables direct intracellular detection of phospho-specific interactions. At the heart of this assay is a chimeric substrate protein that was created by fusing ERK2, a member of the MAPK family and a model post-translationally modified protein[11], with the reporter enzyme TEM-1 Bla. To validate the assay, *E. coli* cells were transformed with plasmids encoding ERK2-Bla, MEK1[R4F], a constitutively active mutant of the upstream activating kinase that can be functionally expressed in the cytoplasm of *E. coli*[35], and different phospho-specific DARPins that distinguished between the nonphosphorylated or doubly phosphorylated forms of ERK2[30]. Upon co-expression of these three key components in *E. coli*, the PhLI-TRAP assay reliably reported the specificity and selectivity of five different DARPins. The utility of PhLI-TRAP was then confirmed by implementing a high-throughput selection process that enabled: (i) affinity maturation of the pE59 DARPin for its cognate pERK2

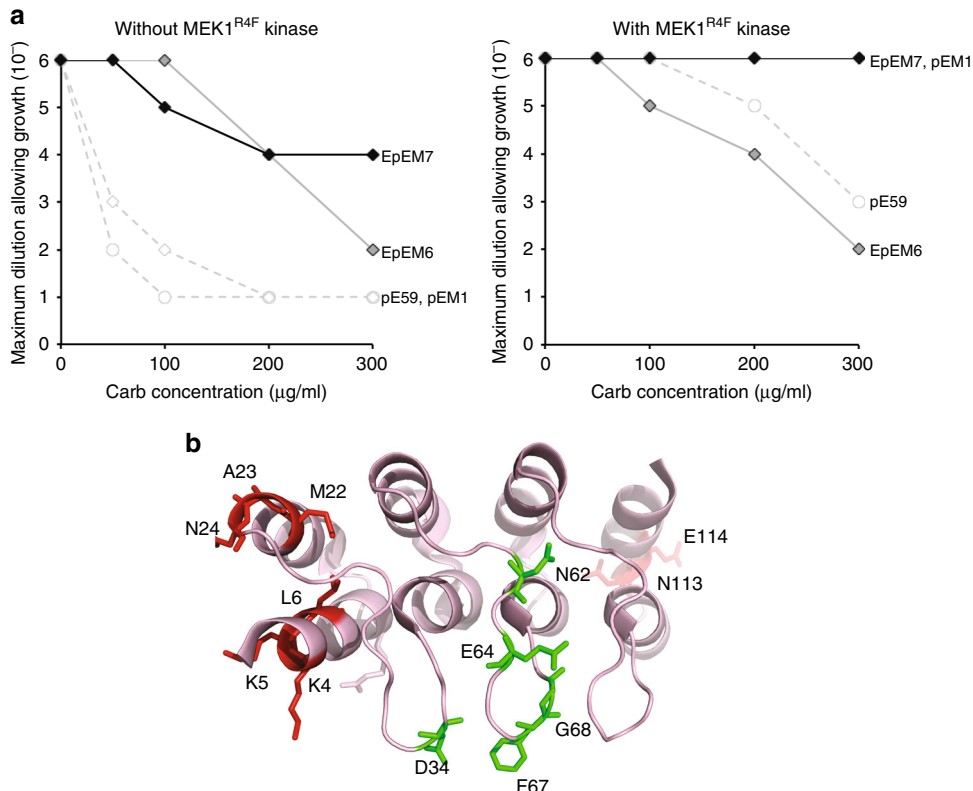

**Fig. 5** Isolation of DARPin variants with altered ERK2 specificity. **a** Survival curves for serially diluted *E. coli* MC4100 cells co-expressing TatABC along with ERK2-Bla and Tat-targeted DARPins EpEM6 (dark gray diamonds) and EpEM7 (black diamonds). Resistance of cells was evaluated in the presence (left) or absence (right) of MEK1^R4F kinase as indicated. Overnight cultures were serially diluted in liquid LB and plated on LB agar supplemented with Carb. Maximal cell dilution that allowed growth is plotted versus Carb concentration. Resistance profiles for pERK2-specific DARPins pE59 (empty circles) and pEM1 (empty diamonds) are shown with (left) and without (right) MEK1^R4F for comparison. **b** Location of mutations in DARPin variants EpEM6 and EpEM7 mapped onto the pE59-pERK2 co-crystal structure. The structure was derived from PDB ID 3ZUV described in Kummer et al.[30], and the schematic was generated using PyMOL software. The DARPin is shown in light red, and the mutations found in helix and loop secondary structures in red and green, respectively. Source data for this figure is available in the Source Data File

antigen; and (ii) reprogramming of the pE59 DARPin to specifically recognize non-cognate ERK2 antigen. Importantly, the ability to uncover phospho-specific DARPin variants exhibiting superior traits simply by demanding bacterial growth on high concentrations of antibiotic, without the need for purification or immobilization of the phosphoprotein target, ensures that our approach is extremely straightforward, especially compared to conventional methods such as animal immunization and phage display[23,25–27]. For example, in vitro display methods such as mRNA, phage, or ribosome display all require the additional steps of immobilizing a purified antigen followed by biopanning, which are more technically demanding and expensive compared to the survival selection of bacterial cells. Moreover, the ability of our approach to rapidly furnish genetically encoded tools for cell biology is significant given the ever-increasing number of known phosphorylation sites and the global phosphorylation changes that are known to occur during disease[4–8].

Our results unequivocally established PhLI-TRAP selection as a viable route to engineering substantial affinity enhancement towards both cognate and non-cognate antigens. In the latter case, we isolated a clone, EpEM6, that acquired enhanced binding affinity for a poor antigen (e.g. EpEM6) while retaining parental binding activity to the cognate antigen, thereby transforming the pERK2-biased DARPin pE59 into a promiscuous binder that now recognized both ERK2 forms. This outcome was reminiscent of results from numerous enzyme engineering studies using in vitro screening techniques in which the evolution of variants is often

met with the acquisition of increased fitness towards the poor/ new function without impairment of the original function[38–40]. As a result, laboratory evolved enzymes typically display a higher degree of promiscuity compared to their parental counterparts. Several groups have shown that this promiscuity can potentially be avoided by implementing both positive selection for the desired trait in combination with negative selection to partially if not completely suppress the original or any other unwanted activities[40–42]. In one notable example involving Cre recombinase, the use of positive screening alone resulted in relaxation of substrate specificity, whereas the combined use of positive and negative screening resulted in switched specificity[40]. While not directly demonstrated here, a genetic counterselection scheme using PhLI-TRAP could be implemented to isolate DARPins with exquisite selectivity. For example, a tightly controlled, inducible promoter could be used to toggle the expression of the target-modifying kinase such that bacterial growth on plates in the presence of inducer could be used to select for library members that bind to phospho-modified antigen after which counterselection of positively selected hits on plates lacking inducer could be used to easily weed out clones that bind to unmodified antigen (or vice versa). Indeed, introduction of the $P_{tac}$ promoter upstream of MEK1^R4F enabled facile discrimination of DARPin selectivity in the presence and absence of the inducer isopropyl β-D-1-thiogalactopyranoside (IPTG) (Supplementary Fig. 8). Another possible permutation of our selection strategy that could be implemented in the future is the co-expression of free

competitor antigen to minimize the possibility that clones having a sequence-based expression advantage would outcompete other binders having higher affinities and/or specificities.

Beyond the identification of PTM-directed binding proteins and their subsequent engineering, we also envision other ways of applying PhLI-TRAP in the future. These opportunities arise from the linkage between bacterial cell resistance and three system components: the binding protein, the post-translational modifying enzyme(s), and the substrate protein. For example, one could imagine using PhLI-TRAP for high-throughput selection of synthetic libraries encoding one of these components to reveal sequence determinants that govern the activity of the post-translational modifying or that define the modified sites on a target protein. It is also conceivable that the genetic selection strategy could be reconfigured for other types of PTMs, in particular those that have been functionally reconstituted in the cytoplasm of living *E. coli* cells, such as *N*-acetylation[43], glycosylation[44–47], neddylation[48], sumoylation[49,50], and ubiquitination[51,52].

## Methods

**Strains and growth selection conditions**. Wild-type *E. coli* strain MC4100 was used for all growth selection experiments. MC4100 cells were co-transformed with plasmid pDD322-TatABC::ERK2-Bla, which included the genes encoding *E. coli* TatABC for increasing the copy number of Tat translocases and the gene encoding the chimeric reporter construct ERK2-Bla, and either plasmid pDD18-spTorA-RGS-6xHis-pE59 or pDD18-spTorA-RGS-6xHis-pE59::MEK1$^{R4F}$ which included the gene encoding MEK1$^{R4F}$. Plasmid pDD322-TatABC::ERK2-Bla was constructed by first PCR amplifying the gene encoding ERK2 and inserting the PCR-amplified gene into plasmid pDD322-TatABC::α-syn(A53T)-Bla[33] in place of the gene encoding α-syn(A53T). Plasmid pDD18-spTorA-RGS-6xHis-pE59 was constructed by replacing the DNA encoding NAC32-FLAG in pDD18-ssTorA-NAC32-FLAG[33] with DNA encoding RGS-6xHis-pE59, which was generated by PCR using plasmid pRDV-pE59[30] as template. The resulting plasmid was further modified by adding the gene encoding MEK1$^{R4F}$ to create pDD18-spTorA-RGS-6xHis-pE59::MEK1$^{R4F}$. To evaluate cytoplasmic co-expression of ERK2 and MEK1$^{R4F}$, we used plasmid pET-His6-ERK2-MEK1_R4F_coexpression, which was a gift from Melanie Cobb (Addgene plasmid #39212). To evaluate other DARPins, the gene encoding pE59 in each of these plasmids was replaced with PCR products encoding the DARPins E8, E38, EpE82, or EpE89, which were PCR amplified using plasmids pDST67-E8, pDST67-E38, pRDV-EpE82, and pRDV-EpE89 as template DNA[30]. All primers used in the construction of these plasmids are listed in Supplementary Table 2 and all plasmids generated in this study were confirmed by DNA sequencing.

Following transformation, bacteria were grown overnight at 37 °C in Luria Bertani (LB) medium supplemented with 25 µg/ml chloramphenicol (Cm) and 10 µg/ml tetracycline (Tet). The next day, antibiotic resistance of bacteria was evaluated by spot plating 5 ml of serially diluted overnight cells that had been normalized in fresh LB to OD$_{600}$ = 2.5 onto LB agar plates supplemented with 1.0% arabinose, 25 µg/ml Cm, 25 µg/ml Tet, and varying amounts of Carb (0–600 µg/ml). Plated bacteria were incubated at 30 °C for 48 h. *E. coli* strain XL1-Blue was used for cytoplasmic expression of DARPins from pDST67-based plasmids[30]. Cultures were grown in LB medium supplemented with 50 µg/ml ampicillin (Amp), and protein expression was induced with 1 mM isopropyl β-D-1-thiogalactopyranoside (IPTG). For testing the selection/counterselection strategy based on IPTG-inducible MEK1$^{R4F}$ expression, *E. coli* strain MC4100(DE3) cells were cotransformed with plasmids pDD322-TatABC::ERK2-Bla, pDD18-spTorA-RGS-6xHis-pE59, and pEXT22-MEK1$^{R4F}$. The latter plasmid was constructed by PCR amplifying the gene encoding MEK1$^{R4F}$ and ligating the PCR product into pEXT22. Strain BL21(DE3) was used for cytoplasmic expression of ERK2 and pERK2 from pLK1_ERK2 and pLK1_ERK2 + MEK1R4F plasmids, which introduced N-terminal Avi tags for biotinylation in vivo using pBirAcm (Avidity)[30] and C-terminal 6×-His tags for affinity purification and immunodetection. The Avi tags were used for avidin resin purification, after which 6×-His tags were used for Ni-column purification to remove unbound biotin and enhance purity. Cultures were grown in LB medium supplemented with 50 µg/ml Amp, and when OD$_{600}$ reached ~0.3, IPTG (0.4 mM) and biotin (5 µM) were added for protein induction and biotinylation, respectively.

**Library construction and selection**. A random mutagenesis library was generated from pE59 using the Genemorph II random mutagenesis kit (Stratagene). PCR was performed using 1 ng pDD18-spTorA-RGS-His-pE59::MEK1$^{R4F}$ as template in each reaction. The resulting PCR products were digested by XbaI and SalI, purified by gel electrophoresis, and cloned into pDD18-spTorA-RGS-His-pE59::MEK1$^{R4F}$ that had been digested with the same enzymes. The library was transformed into electrocompetent DH5α cells and selected on LB agar containing Cm to recover clones containing the plasmid. The library size and error rate were determined to be 2 × 10⁶ members and ~3 mutations per gene, respectively. The plasmid library was miniprepped from DH5α and used to transform electrocompetent MC4100 cells already harboring the pDD322-TatABC::ERK2-Bla plasmid. Transformed cells were incubated at 37 °C for 1 h without any antibiotics and then were subcultured into fresh LB containing 25 µg/ml Cm and 10 µg/ml Tet to ensure that cells contained both plasmids. After ~16 h, cells were spun down and normalized in fresh LB to OD$_{600}$ = 2.5 followed by direct plating of 100 µl of diluted cells (to the dilution factor previously determined by spot plating) onto LB agar supplemented with 1% arabinose and 300–500 µg/ml Carb. Hits were randomly picked after incubation at 30 °C for ~48–72 h. An identical selection of cells carrying the pDD18-spTorA-RGS-6xHis-pE59 or spTorA-RGS-6xHis-pE59::MEK1$^{R4F}$ (either with or without co-expression of MEK1$^{R4F}$) was performed as negative control. Randomly chosen positive clones were screened by spot plating to confirm Carb resistance and then sequenced to determine the identity of any mutation(s). After sequencing, the genes encoding the DARPin hits were PCR amplified, back-cloned into pDD18 with and without MEK1R4F, and used for spot plating analysis to confirm binding affinity against pERK2 and ERK2.

**Subcellular fractionation and western blot analysis**. To prepare subcellular fractions for western blot analysis, 50 ml of induced culture was harvested and pelleted after 20 h incubation in 25 °C. Cells were resuspended in 1 ml subcellular fractionation buffer (30 mM Tris–HCl, 1 mM ethylenediaminetetraacetic acid (EDTA), 0.6 M sucrose) and then incubated for 10 min at room temperature. After adding 250 µl of 5 mM MgSO$_4$, cells were incubated for 10 min on ice. Cells were spun down, and the supernatant was taken as the periplasmic fraction. The pellet was resuspended in 250 µl phosphate buffered saline (PBS) and sonicated on ice. Following centrifugation at 18,500 × g for 20 min at 4 °C, the second supernatant was taken as the cytoplasmic soluble fraction, and the pellet was the insoluble fraction. To prepare samples for cell lysate analysis, 25–50 ml of induced culture was pelleted and resuspended in 500 µl Bugbuster Mastermix. Samples were rotated at room temperature and then spun down at 18,500 × g for 20 min at 4 °C. The supernatant was taken as the soluble cytoplasmic fraction. Proteins were separated using Precise Tris-HEPES 4–20% SDS-polyacrylamide gels (Thermo Scientific), and western blotting was performed according to standard protocols. Briefly, proteins were transferred onto polyvinylidene fluoride (PVDF) membranes, and membranes were probed with the following antibodies: rabbit anti-p44/42 MAPK (Erk1/2) antibody (Cell Signaling; cat # 4695 S) at 1/2500 dilution to detect ERK2-Bla fusion; rabbit anti-p-p44/42 MAPK (Erk1/2) (Cell Signaling; cat # 9101 S) at 1/2500 dilution to detect pERK2-Bla fusion; mouse anti-RGS-4xHis (Qiagen; cat # 34610) at 1/2500 dilution to detect DARPins; and rabbit anti-GroEL (Abcam; cat # ab90522) at 1/30,000 dilution to detect GroEL, which served as a fractionation marker.

**Protein purification**. For DARPin purification, bacterial cells were harvested by centrifugation and the resulting cell pellets were resuspended in binding buffer (20 mM sodium phosphate, 500 mM NaCl, 20 mM imidazole, pH 7.4). The cell suspensions were then passed five times through an EmulsiFlex™-C5 cell homogenizer (Avestin; 15,000 psi/4 °C) and centrifuged at 15,000 × g for 30 min at 4 °C. The clarified lysate was filtered through a 0.2-µm-syringe filter prior to sample loading. The sample was initially loaded through a 1-ml Ni-resin (GE Healthcare). The column was then washed with buffer containing 20 mM sodium phosphate, 500 mM NaCl, 60 mM imidazole, pH 7.4. The captured protein was eluted with buffer containing 20 mM sodium phosphate, 500 mM NaCl, 250 mM imidazole, pH 7.4. Final purity of proteins was confirmed by SDS-polyacrylamide gel electrophoresis (PAGE) and Coomassie staining. Purity of all proteins was typically >95%.

For biotinylated ERK2 and pERK2 purification, bacterial cell pellets were harvested by centrifugation, pelleted, and resuspended in PBS (pH 7.4) with 1 mM DTT and 0.05% Tween-20. The cell suspensions were then homogenized as above. The soluble lysate containing biotinylated ERK2 and pERK2 was first purified using avidin agarose (Thermo Scientific). The lysates were then loaded onto the packed-avidin agarose column by gravity flow. The column was washed twice with PBS buffer, after which purified fusion protein was eluted using PBS buffer containing 2 mM biotin. The eluents were passed over a Ni-column to further enhance their purity and to remove unconjugated biotin, and the proteins were eluted with buffer containing 20 mM sodium phosphate, 500 mM NaCl, 250 mM imidazole, pH 7.4. Biotinylated ERK2 and pERK2 were analyzed by SDS-PAGE followed by Coomassie staining to confirm purity, which was typically >95% for both proteins.

**ELISA**. Biotinylated ERK2 and pERK2 (100 nM) were immobilized on neutravidin-coated ELISA plates for 2 h at 4 °C, and then washed twice with PBS (pH 7.4) with 1 mM DTT and 0.05% Tween-20. Next, the plates were blocked with PBS (pH 7.4) with 1 mM DTT, 0.05% Tween-20, and 1% (w/v) BSA. All subsequent ELISA steps were performed at 4 °C in PBS (pH 7.4) with 1 mM DTT and 0.05% Tween-20. To measure binding activity, different concentrations of purified DARPins (pEM1, pEM2, pEM3, and pE59) ranging from 0 to 200 nM were applied to wells with or without ERK2 or pERK2 for 1 h at 4 °C. DARPin binding was detected by mouse anti-RGS-4xHis antibody (Qiagen; cat # 34610) at 1/2500 dilution followed by a

goat anti-mouse IgG-HRP conjugate (Abcam; ab6789) at 1/5000 dilution, both in PBS (pH 7.4) with 0.05% Tween-20. After 1 h of incubation at room temperature, plates were washed and then incubated with SigmaFast OPD HRP substrate (Sigma) for 30 min in the dark. The reaction was quenched with 3 M $H_2SO_4$, and the absorbance of the wells was measured at 492 nm.

**SPR**. Kinetic SPR measurements were made using a Biacore 3000 instrument (GE Healthcare) for DARPins pE59, pEM1, EpEM6, and EpEM7. The running buffer was 50 mM Tris (pH 7.4), 150 mM NaCl, 0.05 mM EDTA, and 0.005% Tween-20. Biotinylated ERK2 or pERK2 was immobilized on a streptavidin SA chip (GE Healthcare) to ~500 response units (RU). Interactions were determined by injecting varying concentrations of each DARPin at a flow rate of 30 μL/min for 5 min, after which off-rate measurements were made by flowing running buffer for 50 min. The signal of an uncoated reference cell was subtracted from the sensorgrams. Zero-concentration samples (Tris buffer) were also included in SPR experiments as a baseline for double referencing. Sensorgram data were evaluated by fitting the equilibrium binding responses to obtain affinity values using BIAevaluation software (GE Healthcare) and Prism software.

**Reporting summary**. Further information on research design is available in the Nature Research Reporting Summary linked to this article.

## Data availability

All data generated or analyzed during this study are included in this article (and its supplementary information) or are available from the corresponding authors on reasonable request. The source data underlying Figs. 2a, 2b, 3, 4a, 4b and 5a are provided as a Source Data file.

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

## Acknowledgements

We thank Dr. Melanie Cobb for plasmid pET-His6-ERK2-MEK1_R4F_coexpression (Addgene plasmid #39212) used in this work. This work was supported by the Defense Threat Reduction Agency (GRANT11631647 to M.P.D.), National Science Foundation (grant # CBET-1605242 to M.P.D.). The work was also supported by seed project funding (to M.P.D.) through the National Institutes of Health-funded Cornell Center on the Physics of Cancer Metabolism (supporting grant 1U54CA210184-01). The content is solely the responsibility of the authors and does not necessarily represent the official views of the National Cancer Institute or the National Institutes of Health. B.M. and D. W.-Z. were each supported by a Royal Thai Government Fellowship. M.B.L. and E.A.S. were each supported by a National Science Foundation Graduate Research Fellowship (grant # DGE-1650441 and DGE-1144153, respectively) and a Cornell Presidential Life Science Fellowship. M.B.L. was also supported by a Cornell Fleming Graduate Scholarship.

## Author contributions

B.M. designed research, performed all research, analyzed all data, and wrote the paper. M.B.L., E.A.S., A.J., H.-C.L., and D.W.-Z. performed research. L.K., F.B., and A.P. aided in data interpretation. M.P.D. directed research, analyzed data, and wrote the paper.

## Additional information

**Competing interests:** The authors declare no competing interests.

