## [Peer Review File · Nature Communications]

Reviewers' Comments:

Reviewer #1:

Remarks to the Author:

In this study, the authors report a genetic selection strategy for the isolation of non-immunoglobulin binding scaffolds against phospho-proteins in E.Coli. They modified the previously described FLI-TRAP assay, which is based on the interaction of a Tat-signal coupled protein with an antibiotic resistance conferring ligand. Upon binding of these two interaction partners in the cytoplasm of the bacteria the complex is exported into the periplasm via Twin-arginine translocation resulting in antibiotic resistance. Exposure of the bacteria to increasing concentrations of antibiotics can therefore be used as selection pressure for efficient protein-protein interaction. Here, the authors went one step further by additionally delivering a kinase (MEK1) that specifically phosphorylates the Tat-signal coupled protein (ERK2) to the bacteria in order to select for binding proteins recognizing phospho-ERK2. They used designed ankyrin repeat protein (DARPin) libraries as antibiotic resistance conferring binding partners in their assay. Specifically, they generated a DARPin library based on a previously selected anti-pERK2 binder (DARPin pE59) by error prone PCR. As a proof of concept, this library was subjected to their new phospho-FLI-TRAP (PhLI-TRAP) assay to identify binders that are superior in affinity and/or specificity for pERK2 compared to pE59.

This is a sophisticated and elegantly performed proof-of-concept study. The manuscript is written very clearly and the story flow can easily be followed. Nevertheless, there are several questionable aspects that need to be considered.

General comments:

1. The title of the manuscript tends to be an overstatement and could therefore be misleading. Clearly the authors proof the feasibility to identify phospho-specific binders with their clever and interesting PhLI-TRAP selection technique. However, they extrapolate these findings to a relatively broad field of post-translational modifications (PTM) including glycosylations, acetylations, methylations etc. From the data provided in this manuscript it is not clear whether selections against other PTMs than phosphorylations will be successful.
2. Another major point is, that the authors started out with a previously identified anti-pERK2 DARPin (pE59) and it is unclear whether it would be possible to select binders against a new PTM-target using an "unbiased" library. What they present in this work rather reflects a technique for affinity/ specificity maturation of pre-existing binders rather than a "selection of synthetic binding proteins" as they claim in the title.
3. The authors have previously described the FLI-TRAP technique for the identification of protein-protein interactions (Waraho et al., 2009, PNAS). It is nice to see that they now further developed this genetic selection strategy for the identification of DARPins in the context of phosphorylated target proteins but the actual advantage of PhLI-TRAP over other selection techniques is questionable. Cloning of all vectors and the generation of the bacterial strains might just be as challenging as performing an ex vivo ribosome display.
4. The efficacy of DARPin production in bacteria may greatly vary between individual binders and is dependent on their sequence. Certain DARPins might even be "toxic" for the bacteria. The question remains whether DARPins that have a sequence-based expression advantage would outcompete other binders that have higher affinities/specificities in this in vivo system.

Specific comments:

1. In Fig. 2 the 100 µg/ml Carb plating pictures seem to be identical to those in Supplementary Fig. 2a. This might be conflicting with data duplication policy.
2. For all selections the actual plating pictures are provided except for Fig. 3. If these results are not shown in the main figure, they should be included in the supplementary figures.

3. Make sure that the affinities in Supplementary Fig. 4C and 6 are correct. "10.5 x 10⁻⁹ M" instead of "10.5 M"
4. Please check the labelling of Supplementary Fig. 4 b sensorgrams. It is unclear which panel reflects pERK2 and ERK2 measurements.
5. Judging from the on- and off-rates in the sensorgrams (Supplementary Fig. 4 and 6) it is hard to believe that the binders have affinities in the (sub-) nanomolar range. Especially off-rates seem enormously high. Please double check this data.

Reviewer #2:

Remarks to the Author:

NCOMMS-18-27762:

Meksiriporn, et al., demonstrate a novel approach for selection of phospho-specific DARPins, using a previously engineered phospho-ERK2-specific DARPIn as a model. Binding reagents to detect the phosphorylation state of target proteins (and post-translational modifications in general) are difficult to develop but greatly in demand, and therefore technologies to ease their development are important. Here, the authors use a cleverly designed tool (FLI-TRAP) that offers clear advantages for such work. While the most challenging target would be developing a phospho-specific reagent either de novo or from an existing non-phosphorylated ERK2 or promiscuous ERK2 binder, the authors convincingly demonstrate the key aspects of the approach by establishing the ability to easily discriminate ERK2 and pERK2 by antibiotic resistance and further by selection of a combinatorial library to both improve affinity and alter (broaden) specificity. Along with their discussion of counterselection (and inclusion of feasibility data in Supp. Fig. 7), the results adequately highlight the potential of the Tat-based system for these applications. The authors' conclusions that the method is effective appear well-justified, and the methods are well-described with one or two minor exceptions noted below. Serious problems are apparent with the quantitative binding analysis by SPR, but the ELISA data of Fig. 4 provides confidence that pERK2-specific DARPins have indeed been obtained as suggested.

A few small corrections need to be made to the writeup; in addition, one major problem with the quantitative data exists and must be addressed.

1. p. 7, line 187: A reference is made to Supplementary Fig. 2c, which does not exist.
2. The methods for library selection should indicate how many randomly-chosen clones were screened and somewhere the authors should indicate the proportion that were false positives and statistics on repeated recovery of isogenic clones.
3. Methods for the Western blot analysis should indicate the concentrations or dilutions for the antibodies used.
4. The description of the SPR analysis on p. 16 refers to a different buffer used for E40. E40 was not a clone analyzed in this work.
5. Lines 501-502 indicate that rate constants and the equilibrium constant were determined by fitting a Langmuir 1:1 binding model (implying that the K_d values were calculated as k_{off}/k_{on}), whereas the caption for Supp. Fig. 4 indicates that the K_d values were determined by fitting the equilibrium binding response. The former seems to be correct since the K_d values in Supp. Fig. 4c appear to match the ratio of the kinetic constants.
6. Simple visual inspection of the SPR data in Supp. Figs. 4 and 6 make it obvious that the values presented for k_{on}, k_{off}, and K_d (mis-labeled with units M instead of nM in both tables) are entirely erroneous. For example, the indicated k_{off} of pEM1 binding to pERK2 would yield a half-life > 10,000 s. The data suggests a half-life closer to 10 s or less. Likewise, the equilibrium binding at the indicated injection concentrations would be at or near saturation in the last several injections if the K_d were indeed 0.15 nM (a figure given in a few places, including the abstract), whereas the response is still noticeably increasing. The equilibrium phase of the curves suggests a K_d closer to 50 nM give or take. Other samples are similarly flawed. Furthermore, the published affinity for pE59 binding to pERK2 is 117 nM (ref. 30), more than two orders of magnitude higher than the value indicated here. It should also be noted that the published pE59 affinity is more or less in

agreement with the trajectory of the corresponding ELISA data in Fig. 4b. The model fits to the SPR data should be shown, but something is clearly wrong here.

Point-by-Point Responses to Reviewer Comments

We thank the editor and reviewers for their thoughtful comments regarding our manuscript. In the text that follows, we have responded to each of these comments in a point-by-point fashion (reviewer comments are in black font while our responses are in blue italic font) and have made corresponding revisions to our manuscript. All revisions have been marked in red font in the revised manuscript. We believe these changes have significantly improved the manuscript and that it is now worthy of publication in Cancer Research.

Reviewer #1 (Remarks to the Author):

In this study, the authors report a genetic selection strategy for the isolation of non-immunoglobulin binding scaffolds against phospho-proteins in E.Coli. They modified the previously described FLI-TRAP assay, which is based on the interaction of a Tat-signal coupled protein with an antibiotic resistance conferring ligand. Upon binding of these two interaction partners in the cytoplasm of the bacteria the complex is exported into the periplasm via Twin-arginine translocation resulting in antibiotic resistance. Exposure of the bacteria to increasing concentrations of antibiotics can therefore be used as selection pressure for efficient protein-protein interaction. Here, the authors went one step further by additionally delivering a kinase (MEK1) that specifically phosphorylates the Tat-signal coupled protein (ERK2) to the bacteria in order to select for binding proteins recognizing phospho-ERK2. They used designed ankyrin repeat protein (DARPin) libraries as antibiotic resistance conferring binding partners in their assay. Specifically, they generated a DARPin library based on a previously selected anti-pERK2 binder (DARPin pE59) by error prone PCR. As a proof of concept, this library was subjected to their new phospho-FLI-TRAP (PhLI-TRAP) assay to identify binders that are superior in affinity and/or specificity for pERK2 compared to pE59.

This is a sophisticated and elegantly performed proof-of-concept study. The manuscript is written very clearly and the story flow can easily be followed. Nevertheless, there are several questionable aspects that need to be considered.

We appreciate the reviewer's positive assessment of our work, and recognition that it is sophisticated and elegant but also clearly written and easy to follow. In the section that follows, we addressed the aspects that were encouraged to consider, and feel that the manuscript is significantly improved based on addressing the points of Reviewer #1.

General comments:

1. The title of the manuscript tends to be an overstatement and could therefore be misleading. Clearly the authors proof the feasibility to identify phospho-specific binders with their clever and interesting PhLI-TRAP selection technique. However, they extrapolate these findings to a relatively broad field of post-translational modifications (PTM) including glycosylations, acetylations, methylations etc. From the data provided in this manuscript it is not clear whether selections against other PTMs than phosphorylations will be successful.

Reviewer #1 raises a fair point about the generality of the results and whether PTMs beyond phosphorylation would be compatible with the FLI-TRAP mechanism. While we are confident that other systems for producing cytoplasmic PTMs in E. coli (e.g., N-acetylation, glycosylation, neddylation, sumoylation, and ubiquitination; see manuscript discussion for specific references) could be adapted for FLI-TRAP, we have not directly shown that in the current manuscript. Hence, we have revised the title to more accurately reflect the results presented here.

2. Another major point is, that the authors started out with a previously identified anti-pERK2 DARPin (pE59) and it is unclear whether it would be possible to select binders against a new PTM-target using an “unbiased” library. What they present in this work rather reflects a technique for affinity/specificity maturation of pre-existing binders rather than a “selection of synthetic binding proteins” as they claim in the title.

Reviewer 1 raises another valid point about the title of our manuscript. To address this comment, we have also revised the title to better reflect how the technique was used in the current manuscript (i.e., as a selection for improving pre-existing binders versus a selection for entirely new binders). However, we would like to point out that we have every confidence that the technique could be similarly used to screen naïve libraries to uncover entirely new binders from scratch. This is something that we are currently exploring as a follow-on study to the work described here.

3. The authors have previously described the FLI-TRAP technique for the identification of protein-protein interactions (Waraho et al., 2009, PNAS). It is nice to see that they now further developed this genetic selection strategy for the identification of DARPins in the context of phosphorylated target proteins but the actual advantage of PhLI-TRAP over other selection techniques is questionable. Cloning of all vectors and the generation of the bacterial strains might just be as challenging as performing an ex vivo ribosome display.

We agree that cloning and strain creation can be time consuming; however, the most time challenging of these steps – library construction – would similarly need to be completed for ribosome display. So this is not where any advantage would be gained for either system. Rather, once libraries are created, the advantage of the PhLI-TRAP system is that it only requires growth of bacteria on selective agar plates, which requires little effort or technical expertise. In contrast, ribosome display requires the additional step of immobilizing a purified antigen, followed by the procedure of biopanning the ribosome particles displaying the library over an immobilized antigen, which are both technically demanding steps especially compared to the survival selection of bacterial cells.

4. The efficacy of DARPin production in bacteria may greatly vary between individual binders and is dependent on their sequence. Certain DARPins might even be “toxic” for the bacteria. The question remains whether DARPins that have a sequence-based expression advantage would outcompete other binders that have higher affinities/specificities in this in vivo system.

The reviewer correctly points out that the PhFLI-TRAP method can, in theory, lead to the isolation of clones that have (1) higher affinities/specificities for the in vivo-expressed antigen, (2) better expression/solubility due to sequence-based changes, or (3) both. However, because all individual library members are plated on selective agar plates, each has an opportunity for survival and thus are not in direct competition with one another. Thus, by using plate-based selection, we favor the isolation of all of these possibilities, which can then be confirmed and teased apart in follow-on studies as we described. If one were to alternatively select “winners” by using a liquid culture growth selection, this could lead to growth competition where one of these phenotypes outcompetes the other; however, this is not how library selections were performed in this work.

Specific comments:

1. In Fig. 2 the 100 µg/ml Carb plating pictures seem to be identical to those in Supplementary Fig. 2a. This might be conflicting with data duplication policy.

To avoid any conflicts with data duplication policies, we have removed the redundant data image from the panel in Supplemental Fig. 2a.

2. For all selections the actual plating pictures are provided except for Fig. 3. If these results are not shown in the main figure, they should be included in the supplementary figures.

We have now added the spot plate images corresponding to Fig. 3 to the supplementary information of our revised manuscript. This has been added as Supplementary Fig. 3.

3. Make sure that the affinities in Supplementary Fig. 4C and 6 are correct. “10.5 x 10⁻⁹ M” instead of “10.5 M”

These values in the Supplementary Figures and in Table 1 have been corrected and the units are now accurate throughout.

4. Please check the labelling of Supplementary Fig. 4 b sensorgrams. It is unclear which panel reflects pERK2 and ERK2 measurements.

The labels in the sensorgrams have now all been corrected.

5. Judging from the on- and off-rates in the sensorgrams (Supplementary Fig. 4 and 6) it is hard to believe that the binders have affinities in the (sub-) nanomolar range. Especially off-rates seem enormously high. Please double check this data.

As correctly pointed out by the reviewer, the K_D values determined originally were improperly calculated. This stemmed from our attempt to globally fit an appropriate binding model to the entire curve. However, after careful consultation with our collaborator on this work, Dr. Andreas Plückthun, we determined that the more effective way to calculate K_D for this particular DARPin-antigen system was by fitting the equilibrium binding response as he and his coworkers did in their original paper describing these DARPins (Kummer et al., 2012 PNAS). The reason for the difficulty with global fitting stems from the fact that this is a very fast equilibrating system, which is rather typical and best illustrated by pEM1 against pERK2. This is so fast that the SPR cannot be resolved. Plotting the plateau levels against DARPin concentration thus is a more appropriate strategy for obtaining K_D values. The results of this analysis and the corresponding K_D values are now presented in the revised Supplementary Figs. 5 and 7. Importantly, these values not only better reflect the corresponding binding curves but are now much more closely aligned with the values determined previously in Kummer et al. (2012).

Reviewer #2 (Remarks to the Author):

Meksiriporn, et al., demonstrate a novel approach for selection of phospho-specific DARPins, using a previously engineered phospho-ERK2-specific DARPin as a model. Binding reagents to detect the phosphorylation state of target proteins (and post-translational modifications in general) are difficult to develop but greatly in demand, and therefore technologies to ease their development are important. Here, the authors use a cleverly designed tool (FLI-TRAP) that offers clear advantages for such work. While the most challenging target would be developing a phospho-specific reagent either de novo or from an existing non-phosphorylated ERK2 or promiscuous ERK2 binder, the authors convincingly demonstrate the key aspects of the approach by establishing the ability to easily discriminate ERK2 and pERK2 by antibiotic resistance and further by selection of a combinatorial library to both improve affinity and alter (broaden) specificity. Along with their discussion of counterselection (and inclusion of feasibility data in Supp. Fig. 7), the results adequately highlight the potential of the Tat-based system for these applications. The authors conclusions that the method is effective appear well-justified, and the methods are well-described with one or two minor exceptions noted below. Serious problems are

apparent with the quantitative binding analysis by SPR, but the ELISA data of Fig. 4 provides confidence that pERK2-specific DARPins have indeed been obtained as suggested.

A few small corrections need to be made to the writeup; in addition, one major problem with the quantitative data exists and must be addressed.

1. p. 7, line 187: A reference is made to Supplementary Fig. 2c, which does not exist.

We have corrected this error in the revised manuscript.

2. The methods for library selection should indicate how many randomly-chosen clones were screened and somewhere the authors should indicate the proportion that were false positives and statistics on repeated recovery of isogenic clones.

We have updated the Results section of our revised manuscript with this information. Specifically, for the affinity maturation experiment, we added a detailed description (starting at top of page 8) stating how many randomly-chosen clones were selected (10), the number of false positives (1) and the number of isogenic clones recovered (1), yielding 8 unique clones that were confirmed to be true positives. Likewise, for the selectivity reprogramming, we added a detailed description to the revised manuscript (starting at bottom of page 9) stating how many randomly-chosen clones were selected (7), the number of false positives (0) and the number of isogenic clones recovered (0), yielding 7 unique clones that were confirmed to be true positives.

3. Methods for the Western blot analysis should indicate the concentrations or dilutions for the antibodies used.

We have added antibody dilution information to the methods section of the revised manuscript.

4. The description of the SPR analysis on p. 16 refers to a different buffer used for E40. E40 was not a clone analyzed in this work.

We have corrected this error in the revised manuscript.

5. Lines 501-502 indicate that rate constants and the equilibrium constant were determined by fitting a Langmuir 1:1 binding model (implying that the K_d values were calculated as k_{off}/k_{on}), whereas the caption for Supp. Fig. 4 indicates that the K_d values were determined by fitting the equilibrium binding response. The former seems to be correct since the K_d values in Supp. Fig. 4c appear to match the ratio of the kinetic constants.

The original caption for Supp. Fig. 4 (renumbered to be Supp. Fig. 5 in the revised manuscript) contained a typo, as our original analysis was performed by fitting a Langmuir 1:1 binding model with the K_d values calculated as k_{off}/k_{on} . However, for reasons that are explained in comment 6 below, we have now re-analyzed all of the binding data by fitting the equilibrium binding responses. This analysis is now shown in Supplementary Fig. 5 (as well as Supplementary Fig. 7) and the caption has been corrected accordingly.

6. Simple visual inspection of the SPR data in Supp. Figs. 4 and 6 make it obvious that the values presented for k_{on} , k_{off} , and K_d (mis-labeled with units M instead of nM in both tables) are entirely erroneous. For example, the indicated k_{off} of pEM1 binding to pERK2 would yield a half-life $> 10,000$ s. The data suggests a half-life closer to 10 s or less. Likewise, the equilibrium binding at the indicated injection concentrations would be at or near saturation in the last several injections if the K_d

were indeed 0.15 nM (a figure given in a few places, including the abstract), whereas the response is still noticeably increasing. The equilibrium phase of the curves suggests a K_D closer to 50 nM give or take. Other samples are similarly flawed. Furthermore, the published affinity for pE59 binding to pERK2 is 117 nM (ref. 30), more than two orders of magnitude higher than the value indicated here. It should also be noted that the published pE59 affinity is more or less in agreement with the trajectory of the corresponding ELISA data in Fig. 4b. The model fits to the SPR data should be shown, but something is clearly wrong here.

As correctly pointed out by the reviewer, the K_D values determined originally were improperly calculated. This stemmed from our attempt to globally fit an appropriate binding model to the entire curve. However, after careful consultation with our collaborator on this work, Dr. Andreas Plückthun, we determined that the more effective way to calculate K_D for this particular DARPin-antigen system was by fitting the equilibrium binding response as he and his coworkers did in their original paper describing these DARPins (Kummer et al., 2012 PNAS). The reason for the difficulty with global fitting stems from the fact that this is a very fast equilibrating system, which is rather typical and best illustrated by pEM1 against pERK2. This is so fast that the SPR cannot be resolved. Plotting the plateau levels against DARPin concentration thus is a more appropriate strategy for obtaining K_D values. The results of this analysis and the corresponding K_D values are now presented in the revised Supplementary Figs. 5 and 7. Importantly, these values not only better reflect the corresponding binding curves but are now much more closely aligned with the values determined previously in Kummer et al. (2012).

Reviewers' Comments:

Reviewer #1:

Remarks to the Author:

The authors have addressed the concerns raised by the reviewers and edited the manuscript accordingly. One important, however, rather minor issue is remaining. As suggested by the reviewers the authors have re-evaluated the binding kinetics of their SPR measurements (using the equilibrium binding response) and included a table summarizing the calculated affinity constants of all DARPins for ERK2 and pERK2 as well as their selectivity. It appears that the values presented in this manuscript for the original pERK2-binding DARPIn pE59 differ quite a bit from the previously published values (Kummer et al., 2012 PNAS). While Kummer et al. describe a >74-fold selectivity of pE59 for pERK2 over ERK2, this manuscript only finds a >7.7-fold difference. The authors should comment on this, provide a possible explanation and if necessary discuss these discrepancies in the manuscript.

Reviewer #2:

Remarks to the Author:

Meksiriporn, et al., present a revised manuscript describing their work adapting the FLI-TRAP system to isolate phosphorylation-sensitive DARPins against ERK. The authors have done an admirable job of addressing each of the concerns noted in the initial reviews, largely via corrections and additional information in the text, as well as by reanalyzing some key SPR data. The resulting revised manuscript now appears to convincingly demonstrate the utility of their method, while clearly and succinctly conveying the results. With respect to points 3 and 4 raised by reviewer #1, I believe the authors have presented valid arguments regarding (#3) the advantage of PhLI-TRAP over alternatives such as ribosome display and (#4) the impact of plate-based selection compared to liquid culture; however, both of these points address questions likely to be shared by other readers, and I feel that the information contained in the authors' reply is both interesting and relevant. Thus, a brief mention of these points in the discussion seems warranted. Aside from this minor issue, this work appears ready for publication.

Point-by-Point Responses to Reviewer Comments

We thank the editor and reviewers for their thoughtful comments regarding our manuscript. In the text that follows, we have responded to each of these comments in a point-by-point fashion (reviewer comments are in black font while our responses are in blue italic font) and have made corresponding revisions to our manuscript. All revisions have been marked in red font in the revised manuscript. We believe these changes have significantly improved the manuscript and that it is now worthy of publication in Cancer Research.

Reviewer #1 (Remarks to the Author):

The authors have addressed the concerns raised by the reviewers and edited the manuscript accordingly. One important, however, rather minor issue is remaining. As suggested by the reviewers the authors have re-evaluated the binding kinetics of their SPR measurements (using the equilibrium binding response) and included a table summarizing the calculated affinity constants of all DARPin for ERK2 and pERK2 as well as their selectivity. It appears that the values presented in this manuscript for the original pERK2-binding DARPin pE59 differ quite a bit from the previously published values (Kummer et al., 2012 PNAS). While Kummer et al. describe a >74-fold selectivity of pE59 for pERK2 over ERK2, this manuscript only finds a >7.7-fold difference. The authors should comment on this, provide a possible explanation and if necessary discuss these discrepancies in the manuscript.

One reason for this discrepancy is due to a slightly weaker binding measured here for the pE59 DARPin against its cognate pERK2 antigen. The reasons for this slightly weaker binding are currently unknown but could simply be inter-laboratory variability resulting from subtle differences in how the specific steps of the SPR analysis were performed. A second reason for this discrepancy is the extremely weak binding for pE59 against non-cognate ERK2. As a result, the apparent selectivities for both pE59 and pEM1 may be even higher because SPR signals for the non-cognate ERK2 form were very low and thus led to an imprecise estimation of K_D (which is why the K_D values in Table 1 and in Supplementary Figure 5 were written as $>3.5 \times 10^{-6}$). This underestimation of K_D was noted in the original Kummer et al. paper and is also discussed in our manuscript on page 8. We also added a sentence to explain that this could account for at least part of the discrepancy between the two reports.

Reviewer #2 (Remarks to the Author):

Meksiriporn, et al., present a revised manuscript describing their work adapting the FLI-TRAP system to isolate phosphorylation-sensitive DARPins against ERK. The authors have done an admirable job of addressing each of the concerns noted in the initial reviews, largely via corrections and additional information in the text, as well as by reanalyzing some key SPR data. The resulting revised manuscript now appears to convincingly demonstrate the utility of their method, while clearly and succinctly conveying the results. With respect to points 3 and 4 raised by reviewer #1, I believe the authors have presented valid arguments regarding (#3) the advantage of PhLI-TRAP over alternatives such as ribosome display and (#4) the impact of plate-based selection compared to liquid culture; however, both of these points address questions likely to be shared by other readers, and I feel that the information contained in the authors' reply is both interesting and relevant. Thus, a brief mention of these points in the discussion seems warranted. Aside from this minor issue, this work appears ready for publication.

We have added some additional text to the Discussion section that addresses these two points.